# Minimal Surgical Manpower for Living Donor Liver Transplantation

**DOI:** 10.3390/jcm11154292

**Published:** 2022-07-24

**Authors:** Seoung Hoon Kim, Jang Ho Park, Byoung Ho An

**Affiliations:** National Cancer Center, Goyang-si 10408, Gyeonggi-do, Korea; 12791@ncc.re.kr (J.H.P.); 70938@ncc.re.kr (B.H.A.)

**Keywords:** living donor liver transplantation, minimal manpower

## Abstract

Background: Living donor liver transplantation (LDLT) is widely performed with good outcomes in the current era of improved surgical techniques. However, few studies have addressed how many human resources are required in the surgery itself. This study aimed to introduce how to perform LDLT with minimal manpower and evaluate the outcomes in adult patients. Methods: The main surgical procedures of donor and recipient operations of LDLT were performed by a single specialist surgeon who led a team of minimal manpower that only included one fellow, one resident, one intern, and three nurses. He also provided postsurgical care and followed up all the patients as a primary care physician. The outcomes were analyzed from the standpoints of the feasibility and acceptability. Results: Between November 2018 and February 2020, a total of 47 patients underwent LDLT. Ten patients had ABO-incompatible donors. The median age of the overall recipients was 57 years old (36–71); 37 patients (78.7%) were male. The MELD score was 10 (6–40), and the main etiologies were hepatic malignancy (38 patients or 80.9%) and liver failure (9 patients or 19.1%). The median age of the overall donors was 34 years old (19–62); 22 patients (46.8%) were male. All the graft types were right liver except for one case of extended right liver with middle hepatic vein. All donors had an uneventful recovery with no complications. There was one intraoperative mortality due to cardiac arrest after reperfusion in one recipient. Hepatic artery thrombosis was developed in 5 (10.6%) recipients. An acute rejection episode occurred in one patient. The median follow-up period for all the patients was 32.9 months (range, 24.7–39.8). Biliary complications were developed in 11 (23.4%) recipients. In total, 7 (15%) patients died, including 1 intraoperative mortality, 5 from cancer recurrence, and 1 from intracranial hemorrhage. The 1-, 2-, and 3-year overall survival rates in the recipient group were 91.5%, 87.2%, and 85.1%, respectively. Conclusions: LDLT with minimal surgical manpower is feasible under the supervision of a single expert surgeon who has the capacity for all the main surgical procedures in both donor and recipient operations without compromising the outcomes in the present era of advanced surgical management.

## 1. Introduction

In liver transplantation, a diverse group of health care professionals performs their own missions as a transplant team that comprises a coordinator, hepatologist, surgeon, anesthesiologist, social worker, psychologist, nutritionist, and pharmacist. All these constituents are considered as the sine qua non of the transplant surgery and its perioperative management.

Especially from the standpoint of the surgical procedure, two surgical teams, each with its own main operator, are conventionally needed for donor and recipient operations, respectively.

The conventional wisdom is also applied to living donor liver transplantation (LDLT), because the donor operation is performed simultaneously with the recipient operation in order to match the time of resection of the diseased liver and implantation of the new liver graft.

However, operating surgeons for LDLT, especially in terms of the number of proficient surgeons, can sometimes be in short supply for various reasons because highly qualified experts are not freely available with an unlimited number of people in a timely manner from the standpoint of human resources.

Therefore, minimally personnelled LDLT can surface as an issue considering cost-effectiveness in the current era of improved surgical techniques and management.

There has been no report on the feasibility, safety, and outcomes of LDLT by one single main operator who orchestrates and performs all the main surgical procedures, with a team of minimal manpower that only includes one fellow, one resident, one intern, and three nurses.

To address this issue, the aims of the present study were to introduce an LDLT team of minimal manpower led by a single expert surgeon who performs all the main procedures in both donor and recipient operations and to investigate the outcomes.

## 2. Methods

### 2.1. Patients and Study Design

This study included consecutive adult patients (18 years or older) who underwent LDLT at the National Cancer Center, Korea between November 2018 and February 2020. All medical records were reviewed retrospectively.

The selection criteria for living donors were previously described [1,2,3]. LDLTs were indicated in patients according to basic principles, which have remained constant [4,5,6,7].

The study variables included basic patient characteristics, operative details, and outcomes. This study was conducted in accordance with the Declaration of Helsinki (as revised in 2013). The donor risk and expected recipient outcome were fully comprehended by the donors, recipients, and their families. They were also informed of the minimal surgical manpower performing LDLT supervised by a single surgeon. All LDLT procedures were evaluated and approved by the Korean Network for Organ Sharing affiliated to the Ministry of Health and Welfare of Korea. This study was approved by the Institutional Review Board of National Cancer Center, Korea, and informed consent was waived because this was a retrospective study (IRB number: NCC2021-0273, approval date: 10 September 2021).

### 2.2. Surgical Procedure

In all patients, the right lobe of the living donor was used. A single in-house surgeon (S.H.K) was responsible for the orchestration and progress of all the procedures on LDLT, including donor and recipient operations, and he performed all the main procedures in both the donor and recipient operations. The transplant surgical team consisted of one surgeon, one fellow, one resident, one intern, and three nurses. The resident and intern rotated every month according to their training schedules.

In the donor surgery, under an upper midline incision above the umbilicus, the right liver was fully mobilized and parenchymal transection was carried out using the hanging maneuver. The technical details of the donor surgery have been specified previously [8,9].

The surgeon (S.H.K), by himself, performed from skin incision to fascia closure, assisted by two licensed practical nurses. Then, the remaining skin and subcutaneous tissue were closed by the two nurses.

After harvesting the liver graft, the surgeon and one nurse performed the bench procedures, which included infusion of histidine-tryptophan-ketoglutarate solution into the graft, reconstruction of the middle hepatic vein or anomalous portal vein branching, and ductoplasty for multiple bile ducts.

In the meantime, the fellow (J.H.P), a physician who completed a surgical residency, started the recipient operation 15–30 min after the donor operation. He was assisted by one resident in the third or second year and one intern in the first year of training after medical school. He opened the abdomen and mobilized both lobes of the liver. At this time of the recipient operation, the surgeon completed the donor hepatectomy and bench procedure and came over to the recipient operating room. He performed the hilar dissection and complete mobilization of the liver from the inferior vena cava.

Following total hepatectomy in the recipient, with continual assistance from the fellow, implantation started with hepatic vein anastomosis. Any sizable (5 mm or larger in diameter) venous branch of the middle hepatic vein or inferior right hepatic vein of the graft was reconstructed. Considering its size and redundancy, the right portal vein of the graft was anastomosed to either the right or main portal vein of the recipient. Following the reperfusion of the implanted graft, hepatic artery anastomosis was performed under microscopic guidance. When the graft had two separate arteries, the dominant artery was reconstructed first, and then, the back-bleeding from the second hepatic artery was taken for an arterial-blood gas test. Depending on the results, the second artery was anastomosed or ligated [10]. The bile duct was reconstructed with end-to-end duct-to-duct anastomosis. After meticulous hemostasis and insertion of three surgical drains by the surgeon, the abdomen was closed by the fellow, one resident, and one nurse.

### 2.3. Anesthesia Management

One attending anesthesiologist who had been highly trained for LDLT was involved in anesthesia management of the patients. Basic intraoperative monitoring included central venous and intraarterial pressure monitoring. The thromboelastography for coagulation monitoring and rapid infusion devices and red cell salvage systems were used. A rapid response laboratory service with rapid turnaround times and blood bank services were available. To decrease bleeding and transfusion requirements, restrictive fluid administration of 0.5 mL/kg/min was maintained during the donor surgery, and a low central venous pressure of less than 5 cm H_2_O was maintained during the recipient operation.

### 2.4. Transfusion Management

Blood products were transfused to treat acute blood loss associated with surgery. The autologous cell salvaged blood was also used to reduce the need for allogenic blood transfusion. The threshold hemoglobin concentration for transfusion was 8 g/dL for general patients and 9 g/dL for those at a higher risk of adverse effects of anemia. Fresh-frozen plasma (FFP) was transfused in patients with active bleeding and an International Normalized Ratio (INR) greater than 2.0. Platelet transfusion was given if the count was below 50,000/µL.

### 2.5. Immunosuppression

For all LDLT recipients, Basiliximab (20 mg) was used as an induction agent during LDLT and on day 4 after surgery, and maintenance immunosuppressants included tacrolimus, corticosteroids, and mycophenolate mofetil. Corticosteroids were gradually tapered off and stopped within 6 months after LDLT. Tacrolimus was initiated on postoperative day 1, with the dose adjusted at a trough concentration of 8–12 ng/mL during the first month and at 5–8 ng/mL thereafter. Mycophenolate mofetil was administered twice daily from postoperative day 2 with dosage adjustment according to adverse effects.

For desensitization in the ABO-incompatible (ABO-I) LDLT patients, rituximab as a single intravenous dose of 300 mg/m^2^ body surface area was given before LDLT. Intravenous immunoglobulin (0.8 g/kg) was administered on postoperative days 1 and 4. No other methods, such as plasmapheresis, splenectomy, or graft local infusion, were used [11].

### 2.6. Thromboprophylaxis

Anticoagulation therapy was maintained during and after LDLT. In the recipient operation, prostaglandin E1 and antithrombin III were administered for 7–14 days after LDLT. Low-dose aspirin therapy (100 mg/day) was initiated on the 14th postoperative day and continued for 1 year in patients whose platelet counts were no less than 90,000/μL. In the donor surgery, heparin (5 IU/kg) was intravenously injected just prior to dividing the hepatic artery.

### 2.7. Infection Prophylaxis

Patients with hepatitis B virus were administered hepatitis B immunoglobulin in addition to oral antiviral drug for prophylaxis following LDLT. Patients with suspicion of hepatitis C virus recurrence were administered pegylated-interferon and ribavirin after confirmation of hepatitis C virus RNA levels and elevated liver enzyme levels. For prophylaxis against other infections, the patients were administered ticarcillin–clavulanate for one week, fluconazole for one month, and trimethoprim-sulfamethoxazole for one year. Cytomegalovirus prophylaxis was not performed routinely. A cytomegalovirus antigenemia assay was performed twice a week until discharge, every week until one month postoperatively, and every two weeks or once a month thereafter.

### 2.8. Follow-Up and Surveillance

Follow-up Doppler ultrasonography was performed daily during the postoperative two weeks to confirm vascular patency. Hepatic artery thrombosis and biliary complication were diagnosed on the basis of clinical presentation, laboratory findings, and imaging, including dynamic computed tomography (CT).

All donors were followed up with routine laboratory tests at 1 month and 4 months after hepatectomy, and then every 6 months. CT was checked at 1 week, 1 month, and 1 year after operation.

Follow-up of recipients after LDLT included CT scans of the abdomen every 3 months during the first 2 years, every 4 months during the third year, and biannually thereafter.

### 2.9. Statistical Analysis

Data are presented as number (%) or median (range: minimum, maximum) unless otherwise specified. Calculations were carried out using computer software SPSS Version 20.0 (SPSS Inc., Chicago, IL, USA). The survival curves were produced using the Kaplan–Meier method.

## 3. Results

### 3.1. Baseline Patient Characteristics

During the study period, isolated LDLT was consecutively performed in a total of 47 patients at the authors’ institution. Of those, ABO-I LDLT was performed in 10 patients without suitable ABO-compatible living donors.

The median age of the overall recipients was 57 years old (range, 36–71); 37 patients (78.7%) were male. The Model for End-Stage Liver Disease (MELD) score was 10 (range, 6–40), and the main etiologies were hepatic malignancy (38 patients or 80.9%) and liver failure (9 patients or 19.1%). The median age of the overall donors was 34 years old (range, 19–62); 22 patients (46.8%) were male. All the graft types were right liver except one case of extended right liver with middle hepatic vein. The baseline demographic and preoperative characteristics of the recipients and living donors and are summarized in Table 1 and Table 2, respectively.

### 3.2. Operative Outcomes

All LDLs were performed without complete inferior vena cava clamping or venovenous bypass. The operative data of recipients and living donors are summarized in Table 3 and Table 4, respectively.

There was one case of intraoperative death following cardiac arrest within 5 min after graft reperfusion. Before the reperfusion, the patient status had been stable without any blood product transfusion. No identifiable causes were suspected, so the cause was classified as intraoperative cardiac arrest due to postreperfusion syndrome [12].

Intraabdominal bleeding occurred in 5 (10.6%) recipients within 1 week after LDLT, which was treated with reoperation in 3 patients and with angiographic embolization in 2 patients.

Hepatic artery thrombosis developed in 5 (10.6%) recipients, all of which were categorized as early hepatic artery thrombosis, which is defined as any thrombosis within 30 days of transplantation [13]. It was treated with angiographic thrombolysis in four patients and reoperation was required in one patient. A 51-year-old man underwent acute thrombotic hepatic artery occlusion on postoperative day 16. Catheter-directed thrombolysis failed to recover artery flow. Additionally, the attempted stenting caused iatrogenic intimal dissection up to the graft intraparenchymal artery. There was no donor candidate for re-transplantation. Therefore, a right gastroepiploic arteriovenous shunt was created as a salvage treatment for partial portal vein arterialization. The patient is currently doing well and is in good health at two years postoperative [14].

Small-for-size syndrome defined by Dahm et al. [15] occurred in 8 (17%) patients, and all of them were successfully managed without interventions and demonstrated normalization of liver function by 1 month.

An acute rejection episode occurred in one (2.1%) recipient on postoperative day 22, which was steroid-resistant but successfully treated with antithymocyte globulin.

All the recipients except for the one patient of intraoperative mortality were discharged with recovery. The median hospital stay after LDLT of the overall recipients was 16 days (range, 14–86)

All living donors were transferred to the recovery room after being extubated in the operating theatre without being admitted to the intensive care unit. They were discharged without any complication.

### 3.3. Complications and Outcomes after LDLT

The median follow-up period for all the patients was 32.9 months (range, 24.7–39.8). Biliary complications developed in 11 (23.4%) patients (8 stricture, 3 leakage) (Figure 1), which were managed with percutaneous transhepatic biliary drainage.

In total, 7 (15%) patients passed away: 5 from cancer recurrence, 1 from intracranial hemorrhage in head trauma, and 1 from postreperfusion syndrome. The 1-, 2-, and 3-year overall survival rates in the recipient group were 91.5%, 87.2%, and 85.1%, respectively (Figure 2).

All living donors were followed up in the outpatient clinic after discharge. They showed no adverse sequelae not only on the general health status but also on routine laboratory tests and computed tomography scans at 1 month after operation, then 3 months later, and thereafter every half year. They resumed their previous activities with full recovery.

## 4. Discussion

To the best of our knowledge, this is the first study to show the feasibility and safety of LDLT with a single proficient surgeon performing all the main surgical procedures of both the donor and recipient operations under a team of minimal manpower, including only one fellow, one resident, one intern, and three nurses playing the role of physician assistant. Actually, there has also been no report on both the donor and recipient operations performed by one single surgeon in other solid organ transplantation, including kidney, intestines, heart, lung, and pancreas.

Our institution had about a 500-bed-capacity hospital. Since January 2005 when the first LDLT was initiated until this study began, more than 700 LDLTs have been performed by two in-house surgeons, two or three fellows, and changing but not insufficient numbers of residents and interns in the usual manner such as in other LDLT centers. In the meantime, as surgical techniques and postoperative managements have continued to advance, the outcomes of LDLT have continued to improve. However, a big challenge is the recruitment and retention of trained medical personnel such as surgeons, fellows, and residents due to moving to other hospitals and the difficulty in recruiting new members. Despite these negative situations, we have accumulated experience and confidence so that the LDLT can be continued for the sake of the institution’s mission of providing medical care services for cancer patients. This is how this study was initiated. Actually, the surgeon did not have any willingness to perform LDLT under this shortage of manpower, but the circumstances of the institution and transplant team encouraged us to make our own way to perform this type of LDLT.

The study results indicated that the LDLT outcomes were not compromised even with a transplant team with apparently inadequate surgical manpower. No donors showed any complications so far during the regular follow-up after right hepatectomy. In the recipients, one case of intraoperative mortality was encountered following cardiac arrest due to postreperfusion syndrome without any identifiable causes suspected, which is considered an unpredictable and unavoidable complication that was caused not by medical malpractice but by unforeseeable circumstances. One study showed that the incidence of intraoperative cardiac arrest in adult cadaveric liver transplantation was 5.5% with an intraoperative mortality rate of 29.4% [16].

Other major complications such as bleeding, hepatic artery thrombosis, acute rejection, etc. also occurred.

Intraabdominal bleeding occurred in 5 (10.6%) recipients after LDLT. Despite the refinements that have been achieved in medical and surgical techniques, coagulopathy after LDLT may be inevitable for a certain period of time in most liver transplant recipients, until the graft’s function is normalized. There have been only a few published data about post-LDLT hemorrhage. A report from a high-volume center showed that 42 sessions of conventional arteriography were performed in 32 (16.4%) of the 195 patients who underwent LDLT in search of bleeding foci of arterial origin [17]. The A2ALL Consortium reported that 7% of patients who underwent LDLT experienced intra-abdominal bleeding [18]. Another report showed that 18 (15.3%) developed intra-abdominal hemorrhage in 118 adult LDLT patients [19]. Although the incidence and mortality rate of bleeding have not been precisely determined, arterial bleeding after liver transplantation has sporadically been reported to be an obvious cause of death [20,21,22]. In this study, all bleeding cases were treated with reoperation in three patients and with angiographic embolization in two patients; all of them were discharged with full recovery.

Hepatic artery thrombosis after liver transplantation is a lethal complication with an incidence ranging from 2.5% to 15.0% [13,23,24,25]. Our results showed that hepatic artery thrombosis developed in 5 (10.6%) recipients, which could be higher than those of other superior centers. The real etiology of hepatic artery thrombosis remains a matter of debate and is in most cases unidentifiable. Surgical technique was traditionally proposed to be the most important risk factor [26]. However, other risk factors such as graft preservation, ischemia-reperfusion injury, immunological factors, coagulation abnormalities, infections, elderly donors, rejection episodes, retransplantation, prolonged operation time, low recipient weight, and genetic factors could also be implicated [27]. Therefore, it is difficult to say that these complications were entirely due to the factor of minimal surgical manpower. All five cases were managed successfully, although hepatic artery thrombosis represents a major cause of graft loss and mortality after LDLT.

Biliary complications developed in 11 (23.4%) recipients. Studies have reported a 5–40.6% overall incidence of biliary complications after LDLT [28,29,30]. The causes include multiple operative factors such as biliary ischemia, cold ischemia time, type of anastomosis (duct to duct vs. hepaticojejunostomy), single vs. multiple duct anastomosis, surgical expertise, prior bile leak, and donor factors such as age, gender, weight, blood type, and liver steatosis. However, this study could not pinpoint the real etiology because of the small sample size. Further studies with a large sample size need to be carried out for an analysis of this risk factor.

There were many complications in this study. However, all the complications were overcome so that all the recipients except for the one patient of intraoperative mortality were discharged with recovery. Therefore, it is possible to say that these complications could be successfully managed even with minimal surgical manpower. If one factor had to be chosen for these good outcomes, it would possibly be the surgeon’s over 15 years of abundant experience in more than 700 LDLTs and 1000 cases of liver surgery and his proficiency in the surgical details of the donor and recipient operations of LDLT. This may be regarded as one of the critical factors driving LDLT with minimal surgical manpower in this low-volume center, where the operating surgeons for LDLT can sometimes be in short supply.

The ischemic time of the liver graft is a major concern in this study, as it is highly likely to be long because the single main surgeon performed all the main procedures in the donor hepatectomy, bench procedure, recipient hepatectomy, and graft implantation. In LDLT, the surgeon completed the donor hepatectomy and bench procedure with three nurses. Then, he came over to the recipient operating room, and, with the assistance of the fellow, finished the recipient total hepatectomy. The longest cold and total ischemic times were 150 and 172 min, respectively. Warm ischemia is a term used to describe ischemia of the liver graft under normothermic conditions. In the LDLT setting, this term is used to describe two physiologically distinct periods of ischemia: first warm ischemia during organ retrieval, from the time of vascular cross clamping until cold perfusion is commenced, and second warm ischemia during implantation, from the removal of the organ out of ice until reperfusion. The warm ischemic time usually reflects the second warm ischemic time during which inflow and outflow vascular anastomoses by surgeons are performed because the first warm ischemic time usually takes less than 2 or 3 min. In high-volume centers, the warm ischemic time is around 30 to 50 min [31,32,33]. Therefore, the warm ischemic time of 19 min in this study was not considered high even compared with other centers. As a result, no patients showed hepatic injury translated by an elevation in serum ALT and total bilirubin levels. Therefore, ischemic time may no longer be a worrisome feature in this type of LDLT with minimal surgical manpower.

The performance of safe and fast donor hepatectomy is a key success factor to the good outcomes of minimal-manpower LDLT. There were several reports that living donor morbidity and mortality may be inevitable even in the most experienced hands despite every precaution [34,35,36]. However, in this study, there have been no complication or readmission in donors so far. The donor operation time showed a wide difference, where the shortest was 108 min and the longest was 224 min. The reason for the long operation time was twofold: firstly, the surgeon in the middle of the donor operation was sometimes called in to the recipient surgery room due to unexpected problems such as bleeding from vena cava that the fellow could not control, and secondly, some living donors had intra-abdominal adhesion [3].

The longest postoperative hospital stay of 86 days in a recipient may deserve particular mention. A 47-year-old female with hepatic neuroendocrine tumor that had been judged unresectable at other hospitals presented with progressive abdominal distension. She was completely confined to bed and ultimately developed liver failure. Considering the patient’s rapidly deteriorating general condition, LDLT was performed with a right liver from her 21-year-old daughter. The removed liver weighed 14.5 kg while her body weight was 53.9 kg [37].

Efficiency may be translated into the number of resources required to achieve a certain goal. If many resources are used to reach this goal, efficiency is low. If only few resources are used, efficiency is high. We have been updating the surgical technique and patient management of LDLT to achieve the best results. So, even with minimal surgical manpower, as this study demonstrated, a single surgeon could master and perform all the surgical techniques required in LDLT with acceptable outcomes. So, this practice led to the utilization of less resources and manpower, without compromising outcomes, which can contribute to the improvement of the cost-effectiveness of performing LDLT.

Doctors, including surgeons, fellows, residents, and interns, work between 40 and 60 h per week. As hospitals run 24 h a day, nurses usually work in 3 shifts a day—8 h each for the day, evening, and night.

The LDLT volume in our institution was 47 cases over 16 months, calculated as 2.9 cases per month. The median operation time was 423 min. So, for the most part, the LDLT was carried out during working hours (8 a.m. to 5 p.m.). However, when the surgery required longer hours to complete, a surgeon’s shifts were often longer than the standard eight-hour workday and the surgical team worked until the end and took a day off the next day or another. The surgeon or the fellow was always ready to provide consultation over the phone or commute to work to perform patient care, including emergency surgery. The two also took turns going on vacation, choosing a week without surgery.

Surgeon’s fatigue from overwork can be a matter of concern. The surgeon sometimes feels physically exhausted from the burnout that he experiences from this chronically taxing work. Furthermore, he feels emotional exhaustion and low personal accomplishment when the outcomes are unsatisfactory. However, with time and experience, the surgeon can master each and every procedure of LDLT and perform it safely and efficiently with minimal surgical assistance. Therefore, contrary to initial expectations, he felt more personal growth and development rather than being exhausted. Furthermore, the persistence and determination the surgeon possessed was increased because the ability of LDLT to save lives provided him with a very rewarding and professionally challenging surgical career. Fortunately, the surgeon was not so sick that he could not perform LDLT during the study period. Furthermore, the mean monthly volume of LDLT was small. So, the surgery could have been delayed a few days later if the surgeon’s condition was not suitable for the operation schedule.

New surgical methods deserve much attention, and no less important are novel approaches applied to the existing surgical practice. There is no one-size-fits-all approach in the performance of LDLT. Surgeons should use their own professional judgment in the practice of this type of LDLT as the guidance contained in this manuscript may not be appropriate for all surgeons or all situations.

Limitations associated with this study that deserve further comments are the retrospective nature, small sample size, and single-arm design from a single institution. Despite these limitations, we believe that this study provides useful information regarding the clinical outcomes of LDLT from the standpoint of one single surgeon’s capacity under shortages in the surgical workforce.

There may be grave concerns about possible adverse events by surgeons who may read the report superficially and attempt the procedure without meticulous preparations. As such, these results should not be interpreted as implying that these good outcomes can always be extrapolated to other transplant centers. Actually, before this study, the surgeon had rich experience in LDLT and liver surgery and, in the meantime, acquired competence in all the surgical details at the center, where adequate manpower and other resources and management pathways were guaranteed in established transplant services.

In conclusion, the overall procedures in both the donor and recipient of LDLT can be performed without compromising the outcomes by a single surgeon whose surgical expertise has been attained through time and experience, providing evidence that LDLT can be considered with minimal surgical manpower in the present era of advanced surgical management.

## Figures and Tables

**Figure 1 jcm-11-04292-f001:**
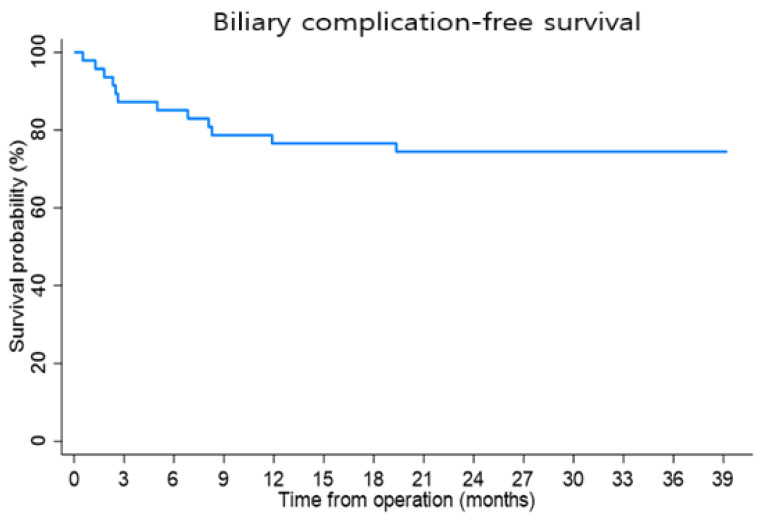
Biliary complication-free patient survival.

**Figure 2 jcm-11-04292-f002:**
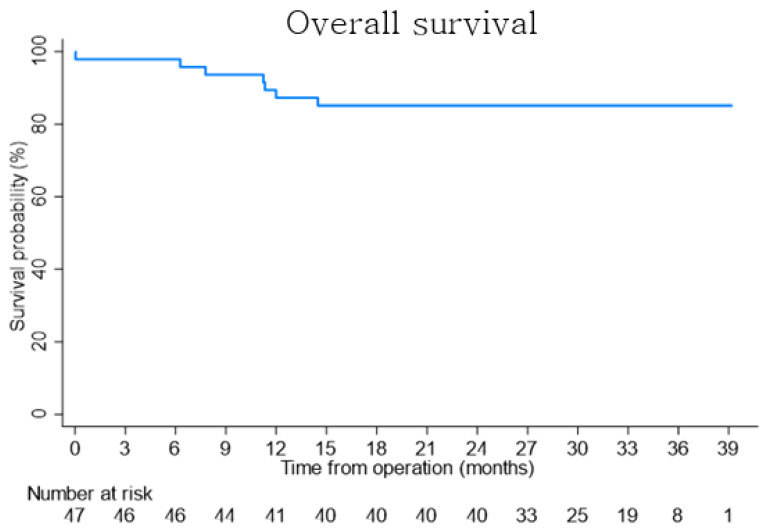
Overall patient survival.

**Table 1 jcm-11-04292-t001:** Baseline characteristics of recipients that underwent LDLT.

	Total (n = 47)
Male (%)	37 (78.7)
Age (years)	57 (36–71)
Body mass index (kg/m^2^)	25.5 (20.14~32.16)
Child-Pugh class	
A	29 (61.7)
B	13 (27.7)
C	5 (10.6)
MELD score	10 (6–40)
Ascites	
None	26 (55.3%)
Mild to moderate	15 (31.9%)
Severe	6 (12.8%)
Graft to recipient weight ratio	0.96 (0.52–1.78)
ABO incompatible	10 (21.3%)
Basal disease	
Liver cancer	38 (80.9%)
Hepatocellular carcinoma	36 (94.8%)
Cholangiocarcinoma	1 (2.6%)
Neuroendocrine tumor	1 (2.6%)
Liver failure	9 (19.1%)
Viral hepatitis type	
B	33 (70.2%)
C	1 (2.1%)
B, C	1 (2.1%)
NBNC	12 (25.6%)
Platelet count (×10^3^/μL)	102 (23–342)

LDLT, living donor liver transplantation; MELD, model for end-stage liver disease. Values are presented as counts with percentages for categorical data and medians with ranges for continuous data.

**Table 2 jcm-11-04292-t002:** Baseline characteristics of donors that underwent LDLT.

	Total (n = 47)
Male (%)	22 (46.8)
Age (years)	34 (19–62)
Body mass index (kg/m^2^)	22.2 (14.6~36.8)
Graft type	
Right liver	46 (97.9)
Extended right liver	1 (2.1)
Remnant liver volume (%)	36.5 (27.2–43.7)
Relationship to recipient	
Child	31 (66)
Spouse	11 (23.4)
Sibling	3 (6.4)
Other relative	2 (4.2)

**Table 3 jcm-11-04292-t003:** Operative outcomes of recipients that underwent LDLT.

	Total (n = 47)
Estimated blood loss (mL)	1500 (100–31,000)
Cold ischemic time (min)	97 (53–150)
Warm ischemic time (min)	19 (7–55)
Total ischemic time (min)	123 (70–172)
Operation time (min)	423 (301–703)
Postoperative hospital stay (days)	16 (14–86)

**Table 4 jcm-11-04292-t004:** Operative outcomes of donors that underwent LDLT.

	Total (n = 47)
Operation time (min)	152 (108–224)
Estimated blood loss	150 (100–400)
Fatty change, macrovesicular (%)	
0~5	38 (80.8)
6~20	6 (12.8)
21~30	3 (6.4)
Postoperative hospital stay (days)	8 (6–13)

## Data Availability

The data that support the findings of this study are available from the corresponding author, S.H.K., upon reasonable request.

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
