# Peer review of "Minimal Surgical Manpower for Living Donor Liver Transplantation"

_jcm, 2022, doi:10.3390/jcm11154292_

Round 1

Reviewer 1 Report

This manuscript was so impressive in terms of perioperative excellent management. I really respect for their efforts. We actually encountered unpredicted events during LDLT, so it was important to collaborate each other. The quality of management usually decreases depending on their busy situation. I was so impressed at their high motivation. These challenging case was feasible by specialized staff, but if one staff will be sick, it might be critical to keep this situation. Anyway I accepted their challenging trial.

Author Response

Our responses(s): Many thanks to your encouraging comments. We appreciate all of your opinions.

The surgeon had many experiences including unpredicted events during LDLT. If one factor had to be chosen, it would possibly be the surgeon’s over 15 years' abundant experience in more than 700 LDLTs and 1000 cases of liver surgery and his proficiency in surgical details of the donor and recipient operations of LDLT. This is considered one of the critical factors driving LDLT with minimal surgical manpower in this 500-bed-capacity hospital.

Fortunately, the surgeon had not been so sick that he couldn’t perform LDLT during the study period. Furthermore, mean monthly volume of LDLT was small (2.9 cases per month). So, the surgery could have been delayed a few days later if the surgeon’s condition had been not suitable for the operation schedule.

Reviewer 2 Report

Dear Authors,

   Thank you for a well written manuscript and communicating with enthusiasm what sheer determination and well-established routines may achieve for peopel in need, such as people in need of a liver graft.

   Transplant surgery is fraught with complications and demand both surgical skills, as well as experience in recognising early adverse effects in addition to establishment of structured pathways in order for these to complications to be dealt with. The authors in this paper succeed both in (re)familiarising us both with the living donor liver transplantation and with its potential complications, as well as proving that a dedicated team, even when the senior surgeon is only one, may achieve great outcomes for patients.

   My only concern would be that while these outcomes are indeed finely presented, one should not think that extrapolating these results in other transplant centers would be always feasible. Transplant surgery remains challenging and, correctly, the authors recognize that possible, not optimal outcomes, may contribute along with physical exhaustion to a burnout, both physical and emotional, of the treating surgeon (-s). 

   The authors' effort in achieving best results for the patients are highly commendable, however I fear that their results are not easily duplicated, and therefore, adequate manpower as well as other resources and management pathways should be guaranteed in established transplant services.  

Author Response

Our responses(s): We appreciate your comments and agree on your concerns. It’s not the intention of this study that the safety can be left solely in the hands of a surgeon and an institution is "compelled" to operate on patients with "apparently inadequate surgical manpower”. As in introduction section, for good outcomes in liver transplantation, a diverse group of health care professionals that comprises coordinator, hepatologist, surgeon, anesthesiologist, social worker, psychologist, nutritionist, pharmacist should perform their own missions as a transplant team. All these constituents are considered as the sine qua non of the transplant surgery and its perioperative management.

However, the operating surgeon for LDLT can sometimes be in short supply especially in low-volume centers. In these circumstances, a proficient surgeon with minimal surgical manpower can contribute to the good outcomes of LDLT. This is the message of this study. 

There may be grave concerns about any possible adverse events somewhere by surgeons who may read the report superficially and attempt the procedure without meticulous preparations.  As such, this result should not be interpreted as implying that these outcomes can always be extrapolated in other transplant centers. Actually, the surgeon had had over 15 years' abundant experience in more than 700 LDLTs and 1000 cases of liver surgery and, in the meantime, acquired his proficiency in surgical details of the donor and recipient operations of LDLT at the center where adequate manpower as well as other resources and management pathways were guaranteed in established transplant services.

Reviewer 3 Report

1. The warm ischemic time 19 minutes is comparatively high. What is the cause of biliary complications were developed in 23.4%. 

2. The incidence rates of intraabdominal bleeding and artery thrombosis are also comparatively high.

3.High incidences of complication due to minimal surgical manpower.

Author Response

Our responses(s): We appreciate your comments.

  1. "Warm ischemia" is a term used to describe ischemia of the liver graft under normothermic conditions. In the LDLT setting, this term is used to describe two physiologically distinct periods of ischemia: first warm ischemia during organ retrieval, from the time of vascular cross clamping until cold perfusion is commenced, and second warm ischemia during implantation, from removal of the organ out of ice until reperfusion. The warm ischemic time usually reflects the second warm ischemic time during which inflow and outflow vascular anastomoses by surgeons are performed because the first warm ischemic time usually takes less than 2 or 3 minutes. In high volume centers, the warm ischemic time was around 30 to 50 minutes.(1-3) Therefore, the warm ischemic time 19 minutes was not comparatively high even compared with other centers.

Biliary complications were developed in 11(23.4%) recipients. Studies have reported a 5%-40.6% overall incidence of biliary complications after LDLT.(4-6) The causes include multiple operative factors such as biliary ischemia, cold ischemia time, type of anastomosis (duct to duct vs hepatico-jejunostomy), single vs double duct anastomosis, surgical expertise, prior bile leak and donor factors such as age, gender, weight, blood type and liver steatosis. However, this study could not pinpoint the real etiology because of a small sample size. Further studies with a large sample size need to be done for the risk factor analysis.

  1. Our results showed that intraabdominal bleeding happened in 5 (10.6%) recipients after LDLT.

In spite of the refinements that have been made in the medical and surgical techniques, coagulopathy after LDLT may be inevitable for a certain period of time in most liver transplant recipients, until the graft's function is normalized. There have been only a few published data about post-LDLT hemorrhage. A report from a high volume center showed that 42 sessions of conventional arteriography were performed in 32(16.4%) of the 195 patients who underwent LDLT in search of bleeding foci of arterial origin.(7) The A2ALL Consortium reported that 7% of patients who underwent LDLT experienced an intra-abdominal bleeding.(8) Another report showed that 18 (15.3%) developed intra-abdominal hemorrhage in 118 adult LDLT patients.(9)

Although the incidence and mortality rate of bleeding have not been precisely determined, arterial bleeding after liver transplantation has sporadically been reported to be an obvious cause of death.(10-12) In our study, all the bleeding cases were treated with reoperation in 3 patients and with angiographic embolization in 2 patients, and all of them discharged with full recovery.

Hepatic artery thrombosis after liver transplantation is a lethal complication with incidence ranging from 2.5% to 15.0%.(13-16) Our results showed that hepatic artery thrombosis was developed in 5 (10.6%) recipients, which could be higher than those of other superior centers.

The real etiology of hepatic artery thrombosis remains a matter of debate and is in most cases, unidentifiable. Surgical technique was traditionally proposed to be the most important risk factor.(17) However, other risk factors such as graft preservation, ischemia-reperfusion injury, immunological factors, coagulation abnormalities, infections, elderly donors, rejection episodes, retransplantation, prolonged operation time, low recipient weight, and genetic factors could also be implicated.(18) Therefore, it would be difficult to say that these complications were entirely due to the factor of minimal surgical manpower.

In this study, all 5 cases were treated successfully though hepatic artery thrombosis represents a major cause of graft loss and mortality after LDLT.

  1. We agree that it can be said that there were many complications in this study. However, all the complications were well overcome so that all the recipients except for the one patient of intraoperative mortality were discharged with recovery. Therefore, it would be possible to say that these complications could be successfully managed even with minimal surgical manpower.

Round 2

Reviewer 3 Report

The paper is generally valuable, but I still have some questions.

1. The secondary surgical hemostasis rate of 10% in the paper is too high. For example, the rate in our center is no more than 2%.

2. 20% rate of biliary complications is also too high, which is 10-15% in our center. Likewise, the incidence of hepatic artery embolism is high.

3. The thermal ischemia time was only 19 minutes. Does this refer to the hepatic free period? The definition of hepatic-free period and thermal ischemia time are different, and the authors' statement may be misleading. Please ask the authors to clarify the definition of thermal ischemia time.

Author Response

This manuscript is a resubmission of an earlier submission. The following is a list of the peer review reports and author responses from that submission.

Round 1

Reviewer 1 Report

I believed perioperative survival rate of living donor liver transplantation depended on the manpower because unpredicted complication might occur during the perioperative time. Not only surgical problem but also the problem associated with patient’s care was also important in terms of quality of life for the recipients including perioperative physical and mental problems. Their clinical outcome was quite impressive, but I have some comments.

Major

1.       One patient was died intraoperatively because of reperfusion injury. Why didn’t they prevent?

2.       How long did their staff including nurse staff work in their hospital?

3.       If the great surgeon’s condition will be sick, how will they respond?

4.       There are a lot of factors related to postoperative complication, but which factor was most important in terms of their excellent outcome?

Minor

1.       What kind of anesthesia management did they do?

2.       What kind of transfusion management did they do perioperatively?

Author Response

Reviewer 1’s Comments to the Author:
I believed perioperative survival rate of living donor liver transplantation depended on the manpower because unpredicted complication might occur during the perioperative time. Not only surgical problem but also the problem associated with patient’s care was also important in terms of quality of life for the recipients including perioperative physical and mental problems. Their clinical outcome was quite impressive, but I have some comments.

Our responses(s): We agree on all of your comments. The followings are the answers to the major and minor comments of the reviewer 1.

Major

  1. One patient was died intraoperatively because of reperfusion injury. Why didn’t they prevent?

Our responses(s): It’s a great pity that one patient in this study died in the middle of surgery. The patient underwent cardiac arrest within 5 minutes after graft reperfusion. Before the reperfusion, the patient status had been stable without any blood product transfusion. No identifiable causes were founded. So, the cause was classified as intraoperative cardiac arrest to postreperfusion syndrome.(1)

Actually, it is considered an unpredictable and unavoidable complication caused not by medical malpractice but by unforeseeable circumstances. One study showed that the incidence of intraoperative cardiac arrest in adult cadaveric LT was 5.5% with an intraoperative mortality rate of 29.4%.(2)

  1. How long did their staff including nurse staff work in their hospital?

Our responses(s): Doctors including surgeon, fellow, resident, and intern work between 40 and 60 hours per week. As hospitals run 24 hours a day, nurses usually work in three shifts a day – eight hours each for the day, evening, and night.

The LDLT volume in our institution was 47 cases for 16 months as calculated 2.9 cases per month. The median operation time was 423 min. So, for the most part, the LDLT was done for the working-hours (8 a.m. to 5 p.m.). However, when the surgery requires longer hours to complete, a surgeon's shifts are often longer than the standard eight-hour workday. The surgeon or the fellow always was ready to provide a consultation over the phone or commute to work to perform patient care including emergency surgery.  

  1. If the great surgeon’s condition will be sick, how will they respond?

Our responses(s): Fortunately, the surgeon had not been so sick that he couldn’t perform LDLT during the study period. Furthermore, the volume of LDLT was not so big (2.9 cases per month). So, the surgery could have been delayed a few days later if the surgeon’s condition had been not suitable for the operation schedule.     

  1. There are a lot of factors related to postoperative complication, but which factor was most important in terms of their excellent outcome?

Our responses(s): If one factor had to be chosen, it would possibly be the surgeon’s over 15 years' abundant experience in LDLT and liver surgery and his proficiency in surgical details of the donor and recipient operations of LDLT. This is considered one big factor driving LDLT with minimal surgical manpower in this 500-bed-capacity hospital.

Minor

  1. What kind of anesthesia management did they do?

Our responses(s): One attending anesthesiologist who had been highly trained for LDLT was involved in anesthesia management of the patients.

Basic intraoperative monitoring included central venous and intraarterial pressure monitoring. The thromboelastography for coagulation monitoring and rapid infusion devices and red cell salvage systems were used. Rapid response laboratory service with rapid turnaround times and blood bank services were available.

All LDLs were performed without complete inferior vena cava clamping or venovenous bypass. This could contribute to more stable hemodynamics during anesthesia.

  1. What kind of transfusion management did they do perioperatively?

Our responses(s): Blood products were transfused to treat acute blood loss associated with surgery.

To decrease bleeding and transfusion requirement, restrictive fluid administration of 0.5ml/kg/min was maintained in donor surgery, and a low central venous pressure of less than 5 cm H2O was maintained in recipient operation. The autologous cell salvaged blood was also used to reduce the need for allogenic blood transfusion.

A threshold hemoglobin concentration for transfusion was 8 g/dL for general patients and 9 g/dL for those at a higher risk of adverse effects of anemia. 

Fresh-frozen plasma (FFP) was transfused in patients with active bleeding and an International Normalized Ratio (INR) greater than 2.0.

Platelet transfusion was given if the count is below 50,000/µL.

Reviewer 2 Report

a well written but unconvincing argument that safety can be left solely in the hands of a surgeon and an institution that is "compelled" to operate on patients with "apparently inadequate surgical manpower"

I would be interested to know the working hours of the various team members particularly during the days when live donor liver transplants were performed.

I also got the impression that this surgeon oversaw the ongoing medical care for these patients.  while they may be able to electively schedule LDLT and allow the surgeon and his team vacation time and hopefully days off service each week, how do they cover emergency complications following surgery and outpatient care such as immunosuppressive management, etc.?

The surgeon may be proud of his feat of endurance, but how long can this be sustained before patients and staff suffer?

Another concern is that 14 of the 17 references are self-citations.

Author Response

Reviewer 2’s Comments to the Author:

a well written but unconvincing argument that safety can be left solely in the hands of a surgeon and an institution that is "compelled" to operate on patients with "apparently inadequate surgical manpower"
Our responses(s): We appreciate your comments. It’s not the intention of this study that the safety can be left solely in the hands of a surgeon and an institution is "compelled" to operate on patients with "apparently inadequate surgical manpower”. As in introduction section, for good outcomes in liver transplantation, a diverse group of health care professionals that comprises coordinator, hepatologist, surgeon, anesthesiologist, social worker, psychologist, nutritionist, pharmacist should perform their own missions as a transplant team. All these constituents are considered as the sine qua non of the transplant surgery and its perioperative management.

However, the operating surgeon for LDLT can sometimes be in short supply especially in low-volume centers. In these circumstances, a proficient surgeon with minimal surgical manpower can contribute to the good outcomes of LDLT. This is the message of this study. 

I would be interested to know the working hours of the various team members particularly during the days when live donor liver transplants were performed.

Our responses(s): Doctors including surgeon, fellow, resident, and intern work between 40 and 60 hours per week. As our hospital runs 24 hours a day, nurses usually work in three shifts a day – eight hours each for the day, evening, and night.

The LDLT volume in our institution was 47 cases for 16 months as calculated 2.9 cases per month. The median operation time was 423 min. So, for the most part, the LDLT was done for the working-hours (8 a.m. to 5 p.m.). However, when the surgery requires longer hours to complete, a surgeon's shifts are often longer than the standard eight-hour workday. The surgeon or the fellow always was ready to provide a consultation over the phone or commute to work to perform patient care including emergency surgery. 

I also got the impression that this surgeon oversaw the ongoing medical care for these patients.  While they may be able to electively schedule LDLT and allow the surgeon and his team vacation time and hopefully days off service each week, how do they cover emergency complications following surgery and outpatient care such as immunosuppressive management, etc.?

Our responses(s): Actually, the surgeon had never been on vacation during the weekday. He only took a short vacation over the weekend staying not so distant from the hospital. The surgeon or the fellow always was ready to provide a consultation over the phone or commute to work to perform patient care including emergency surgery.

The surgeon may be proud of his feat of endurance, but how long can this be sustained before patients and staff suffer?

Our responses(s): Surgeon’s fatigue from overwork can be a matter of concern. The surgeon sometimes feels physically exhausted from the burnout that he experiences from this chronically taxing work. Furthermore, he feels emotional exhaustion and low personal accomplishment when the outcomes are unsatisfactory. However, with time and experience, the surgeon came to get the hang of each and every procedure of LDLT, and perform it safe and efficiently with minimal surgical assistance. Therefore, contrary to initial expectations, he felt more personal growth and development rather than being exhausted. Furthermore, the persistence and determination the surgeon possess was made to be stronger and stronger because the LDLT saving lives provided him with a very rewarding and professionally challenging surgical career.

So, the surgeon hopefully thinks this career will be sustained until physically and mentally possible.   

Another concern is that 14 of the 17 references are self-citations.

Our responses(s): We clearly acknowledge that 14 of the 17 references are self-citations. 

The references # 1-4 are about our selection criteria for living donors that has expanded to include kind of marginal living doors that most LDLT centers were reluctant to operate on.

The references #5-10 give the technical details of donor surgery that were developed and improved by continuous refinement of surgical technique and management.

The reference #11 describes how to manage when the graft had two separate arteries.

The reference #12 shows our unique methods of desensitization in the ABO-incompatible (ABO-I) LDLT patients.

The reference #15 presents right gastroepiploic arteriovenous shunt as a salvage treatment for hepatic artery occlusion after LDLT.

The reference #17 is an LDLT case report that the removed liver weighed 14.5 kg while her body weight was 53.9 kg.

All 14 self-citations described above are our original techniques and methods developed over time and experience in our institution. These have not been reported in other literatures.  

Round 2

Reviewer 2 Report

Thank you for the responses.

Author Response

Thank you for your reviews.

We followed your suggestions faithfully